# Effects of Different Ripening Stages on the Content of the Mineral Elements and Vitamin C of the Fruit Extracts of *Solanum* Species: *S. melanocerasum, S. nigrum, S. villosum,* and *S. retroflexum*

**DOI:** 10.3390/plants13030343

**Published:** 2024-01-23

**Authors:** Jūratė Staveckienė, Brigita Medveckienė, Elvyra Jarienė, Jurgita Kulaitienė

**Affiliations:** Department of Plant Biology and Food Sciences, Vytautas Magnus University Agriculture Academy, 44001 Kaunas, Lithuania; brigita.medveckiene@vdu.lt (B.M.); elvyra.jariene@vdu.lt (E.J.); jurgita.kulaitiene@vdu.lt (J.K.)

**Keywords:** *Solanum*, mineral elements, microelements, macroelements, vitamin C, ripening stage

## Abstract

Studies on the mineral and vitamin C contents of different species and ripening stages of *Solanum* fruits are very limited. The aim of the research was to evaluate the content of the mineral elements and vitamin C of four different *Solanum* species (*S. melanocerasum—*SM, *S. nigrum—*SN, *S. villosum—*SV *and S. retroflexum—*SR), and three ripening stages. The mineral composition of *Solanum* fruits was detected using a CEM MARS 6^®^ (Matthews, NC, USA) digestion system outfitted with a 100 mL Teflon vessel, by microwave-assisted extraction (MAE). In total, eleven mineral elements were detected (K, Ca, Mg, P, Fe, Na, Cu, B, Mn, Al, and Zn). Vitamin C content was assessed by a spectrophotometric method. Depending on the ripening stage/species, content of microelements ranged from 756.48 mg kg^−1^ DW in SV fruits at ripening stage III, to 211.12 mg kg^−1^ DW in SM fruits at ripening stage III. The dominant microelement was Fe. The total content of macroelements in *Solanum* fruits ranged from 26,104.95 mg kg^−1^ DW in SV fruits at ripening stage II to 67,035.23 mg kg^−1^ DW in SR fruits at ripening stage I. The dominant macroelement was K. The data from two experimental years showed that the significantly highest content of vitamin C was in SM fruits and ranged from 48.15 mg 100 g^−1^ at ripening stage I to 45.10 mg 100 g^−1^ at ripening stage III.

## 1. Introduction

*Solanum* sp. are a fruit-producing plant native to Eurasia and introduced in the Americas, Australia, and South Africa, but are a new and promising research subject in the European and Lithuanian agriculture sectors. Among all plant parts, fruits are the most pharmacologically active part [1]. Ripe berries and cooked leaves are edible and are used as food in some locales, and some parts of this plant are used as a traditional medicine. They can also be applied in the food processing industry as natural preservatives and valuable ingredients in developing novel functional products with nutraceutical quality. Moreover, edible fruits of this family are extremely rich in compounds with potent antioxidant activities, such as vitamin C, bioactive glycoalkaloids, glycoproteins, polysaccharides, polyphenolic compounds, and mineral elements beneficial for health [2,3,4]. Mineral elements are scientifically recognized as essential constituents for human health. Vitamin C is crucial for maintaining a healthy body and is involved in over 300 biological processes. It is necessary for the manufacture of collagen [5]. The only natural way humans uptake ascorbic acid is via food. As the fruit enters the ripening phase, there is a slight decrease in vitamin C content. However, in some fruits, the increase in vitamin C continues to the fully mature fruit. The vitamin C content of fruits is never constant but varies with some factors which include climatic/environmental conditions, maturity state, handling and storage, ripening stage, species, and variety of the fruits as well as temperature [6].

We concluded from the literature and experimental result analysis, that *Solanum* sp. fruit can contribute to medical and pharmaceutical practices [7]. Many research studies have proven the various medicinal properties of different parts of the plant. There are studies that show the good nutritional value of *Solanum* fruits. The fruits contain significant amounts of many biologically beneficial components, including 48.20 mg 100 g^−1^ ascorbic acid and 887.70 mg 100 g^−1^ anthocyanins; *Solanum* fruits are therefore ideal raw materials for the manufacture of healthful food. Additionally, fresh *S. nigrum* fruit is rich in protein (2.60%) and has a strong antioxidant activity (229.40 mg 100 g^−1^) [8,9]. Although there are sufficient studies describing the chemical composition of *Solanum* fruits, there is not enough information about how their chemical composition changes during ripening. To the best of our knowledge, there is no information regarding vitamin C and mineral element composition in *Solanum* sp. fruit harvested at different ripening stages.

For these reasons, the aim of this study was to determine the influence of different ripening stages on the content of ascorbic acid, and selected mineral elements (phosphorus, potassium, calcium, boron, natrium, aluminum, manganese, iron, copper, and zinc) content.

## 2. Results

### 2.1. The Effect of the Species and Ripening Stage on the Total Mineral Element Content

Mineral elements are naturally occurring inorganic solid compounds found in many raw materials and are crucial for several physiological processes. One of agriculture’s biggest difficulties is ensuring that people have access to nearly all the organic nutrients and minerals needed to maintain wellness and effective organ operation. People need over 22 mineral elements, some of which are necessary in significant quantities, although other elements like Fe, Zn, Cu, I, and Se are only needed in trace amounts because larger quantities may be harmful [10,11]. 

To select the best period of maturity and use it for nutritional enrichment, it is crucial to understand how the amounts of mineral components change as berries ripen. Our quantitative macroelements and microelements results indicated significant differences between species and ripening stages (Table 1). Depending on the ripening stage/species, the content of microelements ranged from 756.48 mg kg^−1^ DW in SV fruits at ripening stage III to 211.12 mg kg^−1^ DW in SM fruits at ripening stage III. The total content of macroelements in *Solanum* fruits ranged from 26,104.95 mg kg^−1^ DW in SV fruits at ripening stage II to 67,035.23 mg kg^−1^ DW in SR fruits at ripening stage I. Baiyeri et al., 2011 reported that the highest composition of mineral elements in banana fruits was at light-green and/or light-yellow stages (fairly ripe fruits) [12].

### 2.2. The Effect of Species and Ripening Stages on the Content of Macroelements

The determined contents of macroelements demonstrated differences by ripening stage and species. In total, four macroelements were determined: magnesium (Mg), phosphorus (P), potassium (K), and calcium (Ca). K was the primary determined macroelement (Figure 1). The content of this element ranged from 61,204.68 mg kg^−1^ DW in SR fruits at ripening stage I to 23,201.74 mg kg^−1^ DW in SV fruits at ripening stage II. The data demonstrated that K tended to decrease at maturity and the highest contents of potassium were determined to be present in all species of unripe fruits. Potassium concentration in foods of plant origin ranges from 20 to 730 mg 100 g^−1^ fresh weight, while some species like ‘Idaho’ potatoes (*S. tuberosum*) have higher K values. The levels of K in seeds and nuts are considerable, reaching up to 2240 mg 100 g^−1^ [13,14]. Aguirre et al., 2016 determined that this macroelement, dependent on the ripening stage, ranged from 23,095.40 mg kg^−1^ DW to 24,459.90 mg kg^−1^ DW in *Rosa rubiginosa* species [15]. It is crucial to consume the proper quantity of K since this macronutrient aids in the transmission of nerve impulses and helps to support nerve functions by maintaining the balance of the body’s fluid system [13,14,16,17]. 

The second most abundant macroelement is P. Our results show that the content of P ranged from 1399.58 mg kg^−1^ in SM fruits at ripening stage III to 28,874.83 mg kg^−1^ DW in SN fruits at ripening stage II (Figure 1). Data also demonstrated that P tended to decrease at maturity and the highest contents of phosphorus were determined to be in unripe fruits, except SN species. Costa, F. et al., 2011, performed some research into different maturities of *Lycopersicon esculentum* Mill., tomatoes of *Solanaceae* family, and determined that the mineral concentrations of P were higher in red tomatoes than in green ones [18]. Paunovic et al., 2017, determined that the content of P in blackcurrant ranged from 180.00 mg 100 g^−1^ to 199.50 mg 100 g^−1^, depending on different cultivars [19]. In vegetables or fruits, P concentrations range from 16.2–437 mg 100 g^−1^. Fruits have the lowest P content, with a range of 9.9 to 94.3 mg 100 g^−1^. The daily recommended intake for P is 800–1300 mg [20].

The content of Mg in studied *Solanum* fruits ranged from 128.89 mg kg^−1^ in SV species to 1539.93 mg kg^−1^ DW in SR species at ripening stage I. The recommended daily allowance for Mg is 375 mg/day in adults [21]. Other researchers say that in *Solanum nigrum* leaves, the content of Mg ranged from 100–200 mg 100 g^−1^ DW [22]. Sivakumar D et al., 2020, determined in fresh leaves of *S*. *retroflexum* 92 mg 100 g^−1^ DW [23]. Mg is a mineral that is abundant in vegetables and has been linked to important dietary roles in maintaining human health. The content of this macroelement in fruit, berries, and vegetables typically ranges from 5.5 to 191 mg per 100 g of fresh weight, with a daily recommended consumption of 200 to 400 mg [24]. 

The highest content of Ca by a significant margin was established in SN fruits at ripening stage III, 4027.90 mg kg^−1^ DW. Results show that in other species, the content of Ca tended to decrease at maturity and ranged from 1455.13 to 357.25 mg kg^−1^ DW. Involved in the biological processes of numerous tissues, including those of the musculoskeletal, neurological, and cardiac systems, bones, teeth, and the parathyroid gland, calcium is a crucial mineral for maintaining good health in humans. Additionally, Ca participates in the maintenance of mineral homeostasis as well as overall physiological performance and can function as a cofactor in enzyme activities [25,26,27]. 

### 2.3. The Effect of Species and Ripening Stages on the Content of Microelements

There are many elements in natural soil, but only nine are officially regarded as important micronutrients in plants due to their indispensable functions in the entire plant lifecycle. Iron (Fe), zinc (Zn), copper (Cu), manganese (Mn), nickel (Ni), molybdenum (Mo), boron (B), and chlorine (Cl) are all essential for higher plants. However, sodium (Na) is only required for some C4 plants [28,29]. Humans and animals require chromium (Cr), iodine (I), and selenium (Se) in addition to these plant-important microelements. Microelements are utilized in minuscule amounts, accounting for less than 0.1% of dry plant tissue. When taken in large quantities, some particular microelements may be harmful. Except in cases of heavy atmospheric deposition or floods from contaminated waterways, soil is the primary source of microelements for plants. Soil microelements undergo many transformations, and their availability to plants is determined by the chemical form and distribution of soil’s solid and liquid phases, which are regulated by soil conditions, namely pH, texture, and soil aeration status [30].

In *Solanum*, the fruits were determined to contain seven microelements: boron (B), natrium (Na), aluminum (Al), manganese (Mn), iron (Fe), copper (Cu), and zinc (Zn). The most abundant microelement of all these species was Fe (Figure 2). The highest content of this element was detected in SV fruits at ripening stage III 603.83 mg kg^−1^ DW. The results also show that the amount of Fe increases as the fruit ripens. Vegetables and fruits only contain 0.13 to 3.01 mg 100 g^−1^ of F. The majority of the iron in meals with a plant origin is found in insoluble Fe3+ complexes such as phytic acid, phosphates, oxalates, and carbonates. Nevertheless, the bioavailability of Fe in foods is under 8%. Nuts and chocolate powder are good sources of iron (16.1 and 25.8 mg 100 g^−1^, respectively) [21]. The experiment carried out by Otakar Rop et al., 2010, into the different stages of ripening (*Mespilus germanica* L.) shows that the content of Fe increased during ripening [31]. 

Zn concentrations in plant-based foods range from 0.05 to 11.8 mg 100 g^−1^. Fresh fruits have lower levels of Zn (0.02–0.61 mg 100 g^−1^). Low levels of Zn in fruit juices and beverages range from 0.01–0.27 mg 100 g^−1^ [21]. Our results show that the concentration of Zn increased during ripening. The highest content of Zn, 42.41 mg kg^−1^, was detected in SM fruits at ripening stage I. The lowest content of Zn, 17.75 mg kg^−1^ DW, was detected at ripening stage III in SM fruits. Ramesh el al., 2021, conducted research on mineral nutrients in tomato (*Solanum lycopersicum* L.) fruits during ripening. Across varieties and ripening stages, the content of Zn ranged from 17.47µg g^−1^ to 40.35 µg g^−1^ DW. The results show a decreasing tendency in the content of Zn with the progress of ripening [32].

The concentration of Mn ranged from 20.57 mg kg^−1^ DW at ripening stage I in SN fruits, to 35.61 mg kg^−1^ DW at ripening stage III in SM fruits. Additionally, the highest concentration of Cu was detected at ripening stage III (32.84 mg kg^−1^ DW) in SN fruits and the lowest concentration of Cu was at ripening stage II (11.8 mg kg^−1^ DW) in SV fruits. Low manganese (Mn) levels are another characteristic of fruits and vegetables. In both fruits and vegetables, Mn contents have a range of 0.01–0.66 mg 100 g^−1^ and 0.01–0.078 mg 100 g^−1^, respectively [21]. Mn has a daily consumption recommendation of 2 mg, and its primary physiological role is as an enzyme cofactor in antioxidant processes related to glucose metabolism [33,34]. Vegetables have been found to contain low quantities of copper (Cu), ranging from 0.004 to 0.24 mg 100 g^−1^, with the exception of legumes, which can contain up to 0.5 mg 100 g^−1^. Between 0.01 to 0.24 mg 100 g^−1^ of copper can be found in fruits. The recommended daily allowance (RDA) for copper is 1.0 to 1.6 mg per day [21].

Results show the highest content of Na at ripening stage I in SR fruits (88.90 mg kg^−1^ DW). In the Figure 2 we can see that this element has the tendency to increase during ripening. The same tendency is shown with B. Concentration of B varies from 9.57 mg kg^−1^ DW at ripening stage III to 22.13 mg kg^−1^ DW at ripening stage I in SM fruits. The content of Al depends more on the species than on the stage of ripening. The content of Al varies from 10.90 mg kg^−1^ DW at ripening stage II in SM fruits to 40.76 mg kg^−1^ DW at ripening stage III in SV fruits.

### 2.4. The Effect of Species and Ripening Stages on the Content of Vitamin C

The most important vitamin for human nutrition that fruits and vegetables provide is vitamin C [35]. Since humans cannot produce ascorbate, their primary supply of the vitamin is through eating fruit and vegetables. The finest sources of this vitamin are fruits (particularly citrus and tropical fruits). A precise and specific measurement of the nutritional content of fruits is critical for understanding the relationship between dietary intake and human health [36]. Therefore, knowing in which ripening stage the highest amounts of the vitamin are accumulated can help to enrich nutrition. The vitamin C content of several fruits is known to be controlled during fruit growth. The content and concentration of this vitamin in citrus fruit tissues fluctuate depending on the stage of fruit development [37].

In the present study, the data averaged from two experimental years showed that the highest content of vitamin C was in SM fruits and ranged from 48.15 mg 100 g^−1^ at ripening stage I to 45.10 mg 100 g^−1^ at ripening stage III (Figure 3). The results show that vitamin C has a tendency to decrease during ripening in SR, SN, and SM species. The SV species shows the opposite results, vitamin C content increasing as the fruit matures and fluctuating from 31.49 mg 100 g^−1^ at ripening stage I to 38.80 mg 100 g^−1^ at ripening stage II. Data on the vitamin C content of Fresno de la Vega peppers harvested in two different years at different ripening stages (green, red) show values for the content of vitamin C of 106.05 mg 100 g^−1^ and 148.94 mg 100 g^−1^ for green and red peppers, respectively [38]. Mamboleo et al., 2018, determined the content of vitamin C in different stages of maturity in *S. villosum*. The highest relative content of vitamin C was detected at stage of maturation III (85.30 mg 100 g^−1^), and the lowest content at maturation stage I (27.40 mg 100 g^−1^) [39].

### 2.5. Correlation Analysis

The correlation analysis showed that the ripening stage (hue angle) had a very strong relationship with two mineral elements—B (r = 0.998), and Na (r = 0.961)—and a strong positive relationship with four mineral elements—Mg (r = 0.789), P (r = 0.806), K (r = 0.834), and Ca (r = 0.809)—in *S. melanocerasum* species fruits (Figure 4). This demonstrates that mineral element content increased with ripening stage (hue angle) and decreased with ripening.

The study of the *S. nigrum* fruits showed a very strong positive correlation between ripening stage (hue angle) and the mineral elements of B (r = 0.992), Na (r = 0.903), K (r = 0.900), and Mn (r = 0.817), and very strong negative correlation between ripening stage and the mineral elements of Cu (r = 0.933) and Zn (r = 0.945) (Figure 5). This demonstrated that with fruits ripening, these mineral elements increase as well.

Very strong positive correlations were found between ripening stage (hue angle) and vitamin C content of *S. melanocerasum* and *S. nigrum* fruit species (r = 0.976 and r = 0.994, respectively) and a strong correlation was found for *S. retroflexum* fruits (r = 0.730) (Figure 6).

## 3. Materials and Methods

### 3.1. Field Experiment

A two-factor experiment with *Solanum* species (SM—*S. retroflexum*, SN—*S. nigrum*, SV—*S. villosum,* and SM—*S. melanocerasum*), and three different ripening stages—ripening stage I, fruit color green (30% maturity) corresponding to BBCH stage 81; ripening stage II, fruit color 40–60% purplish-violet or yellow-orange (60% maturity), inside incompletely ripe, corresponding to BBCH stage 85; and ripening stage III, fruit color 100% velvety black-blue or orange, inside fully ripe (100% maturity), corresponding to BBCH stage 89 (Figure 7) [40,41]—was conducted during the period 2021–2022 on a farm in the Kaunas district, Lithuania (WGS coordinates 54.8719020, 23.8672686).

### 3.2. Sample Preparation

A combined sample of 1.5 kg for each species and stage of ripening was made after the fruits were collected for analysis. The fruits were washed in tap water, dried, and stored at 34 °C. Using a Freeze-Drying Plant Sublimator 304 05 (ZIRBUS Technology GmbH, Bad Grund, Germany), the samples were lyophilized for 24 h. The fruits were pulverized (Grindomix GM 200, Retsch GmbH, Haan, Germany) after lyophilization and stored at 5 °C in the dark before being analyzed.

### 3.3. Soil Agrochemical Analyses 

An agrochemical auger was used to collect soil samples from the arable layer (0–20 cm depth) in the spring. The soil samples were homogenized, sieved through a 1 mm mesh sieve, and air dried in transparent plastic cases. Agrochemical analyses of the experimental soil were conducted at the Laboratory of Analyses of Vytautas Magnus University Agriculture Academy. The pH, KCl, phosphorus, potassium, and total nitrogen concentrations of soil samples were analyzed. Soil pH was measured according to the potentiometric method using a pH meter in 1 N KCl extract (Lithuanian organization LST) [42]. Using the Egner–Riehm–Domingo method, the ammonium-lactate extraction of the available phosphorus and potassium was performed [43]. The total nitrogen concentration (mg kg^−1^) was determined using the Kjeldahl method. The experimental field soil was characterized by acidity (pH = 4.16), medium potassium status (K_2_O = 78.5–102.2 mg kg^−1^), low phosphorus status (P_2_O_5_ = 45.9–69.3 mg kg^−1^), and 1.25% total nitrogen content.

### 3.4. Mineral Element Analysis 

Using a CEM MARS 6**^®^** (Matthews, NC, USA) digestion system outfitted with a 100 mL Teflon vessel, microwave-assisted extraction (MAE) was used to analyze the mineral composition (K, Ca, Mg, P, Fe, Na, Cu, B, Mn, Al, and Zn) of *Solanum* fruit samples. Approximately 0.3 g of the homogenized sample was accurately weighed into a Teflon vessel and digested using a nitric (HNO_3_) and hydrochloric (HCl) acid mixture (5:1). Digestion was performed under the following conditions: temperature—180 °C; pressure—800 psi; ramp time—20 min; hold time—20 min; microwave power—800 W. Then, the digested sample was then cooled down and thoroughly transferred into a 100 mL volumetric flask and diluted using bi-distilled water to the mark. Each sample was prepared in triplicate and the blank sample was included in each digestion run. Digestion samples were analyzed by means of ICP–MS (ThermoFisher Scientific, Branchburg, NY, USA). 

### 3.5. Vitamin C Analysis

Vitamin C was assessed by a spectrophotometric method which is based on the ability of the ascorbate ion to reduce methyl viologen to a stable blue-colored free radical ion [44]. One gram of plant material was used to create the samples, which were then centrifuged at 2012 g for five minutes after being homogenized with 10 mL of 5% metaphosphoric acid. Then, 1 mL of sample extraction was combined with 2 mL of methyl viologen, 2 mL of 2 M NaOH, and 2 mL of 2 M NaOH. After two minutes, absorbance was measured with a spectrophotometer (M501, Spectronic Camspec, Ltd., Leeds, UK) at a wavelength of 600 nm. The concentration was calculated using the ascorbic acid standard calibration data.

### 3.6. Color Parameter Analysis

*Solanum* fruit color parameters L* (lightness), a* (positive—red, negative—green) and b* (positive—yellow, negative—blue) in NBS units, were evaluated with a spectrophotometer ColorFlex (Hunter Associates Laboratory, Inc., Reston, VA, USA) (Table 2). Chroma (C = (a × 2 + b × 2) 1/2) and hue angle (h° = arctan (b*/a*)) were calculated.

### 3.7. Statistical Analysis 

The data for the *Solanum* sp. mineral elements and vitamin C contents were processed using Microsoft Excel 2016 and the STATISTICA 10 (StatSoft, Inc., Tulsa, OK, USA, 2010) package. The reliability of the results was evaluated using a two-way analysis of variance (ANOVA). The statistical significance of the differences between the means was estimated using Fisher’s LSD test (*p* < 0.05). The relationships between variables were determined using correlation analysis.

## 4. Conclusions

Overall, our findings showed that the amounts of microelements varied between 756.48 mg kg^−1^ DW in SV fruits at ripening stage III and 211.12 mg kg^−1^ DW in SM fruits at ripening stage III, and depended on the ripening stage and species. SV fruits at ripening stage II had a total macroelement concentration of 26,104.95 mg kg^−1^ DW, while SR fruits at ripening stage I had a total macroelement content of 67,035.23 mg kg^−1^ DW. SM fruits had the maximum vitamin C content, ranging from 48.15 mg 100 g^−1^ at ripening stage I to 45.10 mg 100 g^−1^ in ripening stage III.

The results of our investigation verify that the fruits of the studied *Solanum* species are a good source of vitamin C and mineral components. These statistics are useful for selecting the ripening stage of a fruit that will result in the highest concentration of mineral elements and vitamin C. Consequently, it is recommended that fruits from *Solanum* be used in novel food, cosmetics, and pharmaceutical products. More research is necessary to comprehend the mechanism underlying how the ripening stage of fruit affects quality.

Our results showed that the highest amount of microelements was in SV fruits at stage III of ripening, and the highest amount of macroelements was in SR fruits of ripening stage I. The highest amount of vitamin C was determined to be in the SM fruits in ripening stage I.

## Figures and Tables

**Figure 1 plants-13-00343-f001:**
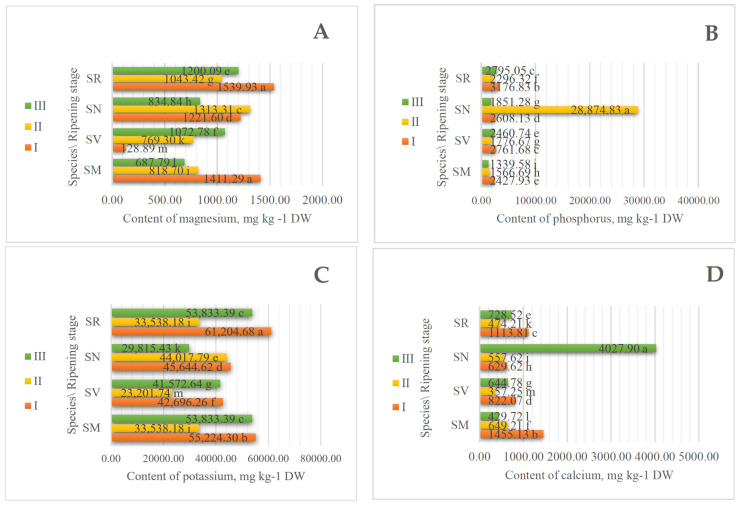
The effect of the ripening stage on the content of macroelements (mg kg^−1^ DW). (**A**) magnesium, (**B**) phosphorus, (**C**) potassium, and (**D**) calcium content of *Solanum* fruits. Different small letters (a–m) represent significant differences between the means (*p* < 0.05); *S. retroflexum*—SR, *S. melanocerasum*—SM, *S. nigrum*—SN, *S. villosum*—SV; ripening stages I, II, and III.

**Figure 2 plants-13-00343-f002:**
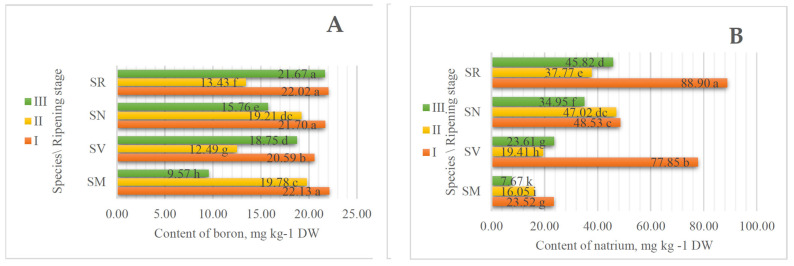
The effect of the ripening stage on the content of microelements (mg kg^−1^ DW). (**A**) boron, (**B**) natrium, (**C**) aluminum, (**D**) manganese, (**E**) iron, (**F**) copper, (**G**) zinc of *Solanum* fruits. Different small letters (a–l) represent significant differences between the means (*p* < 0.05); *S. retroflexum*—SR, *S. melanocerasum*—SM, *S. nigrum*—SN, *S. villosum*—SV; ripening stages I, II, and III.

**Figure 3 plants-13-00343-f003:**
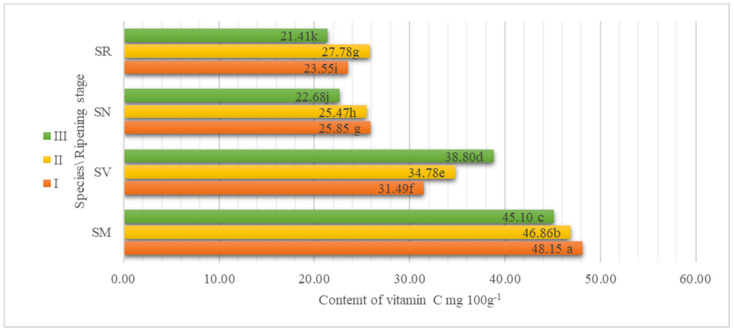
The effect of the ripening stage on the content of vitamin C (mg 100 g^−1^) of *Solanum* fruits. Different small letters (a–k) represent significant differences between the means (*p* < 0.05); *S. retroflexum*—SR, *S. melanocerasum*—SM, *S. nigrum*—SN, *S. villosum*—SV; ripening stages I, II, and III.

**Figure 4 plants-13-00343-f004:**
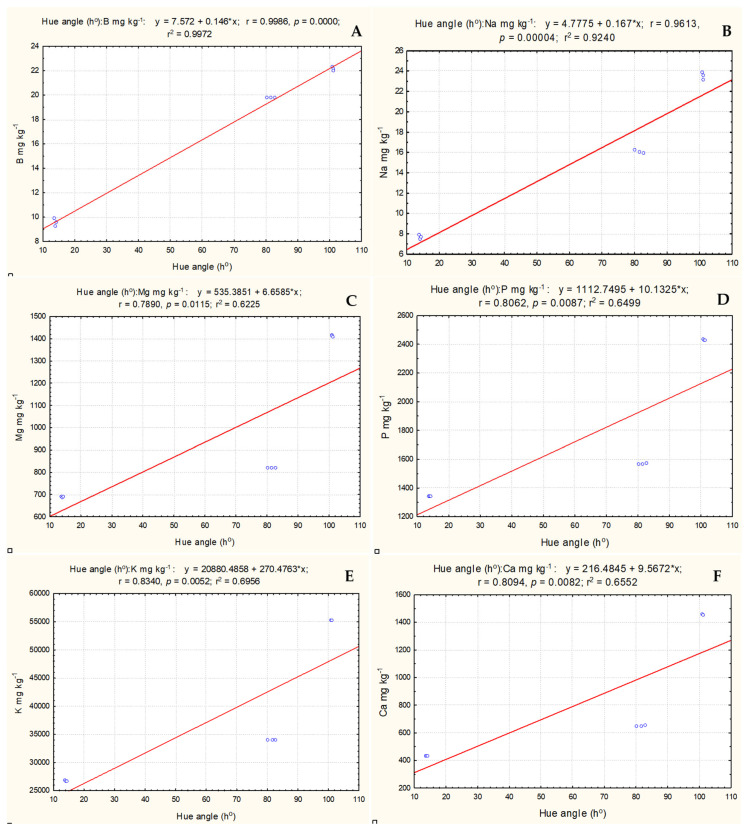
Correlations between ripening stage (hue angle (h°)) and mineral element content. (**A**) borum, (**B**) natrium, (**C**) magnesium, (**D**) phosphorus, (**E**) potassium, (**F**) calcium, (**G**) cuprum, (**H**) zinc.

**Figure 5 plants-13-00343-f005:**
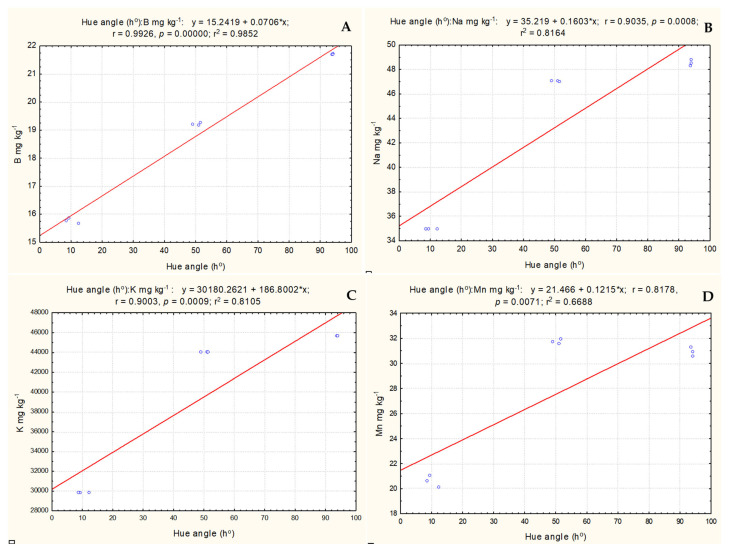
Correlations between ripening stage (hue angle (h°)) and mineral element content. (**A**) borum, (**B**) natrium, (**C**) potassium, (**D**) manganese, (**E**) cuprum, (**F**) zinc.

**Figure 6 plants-13-00343-f006:**
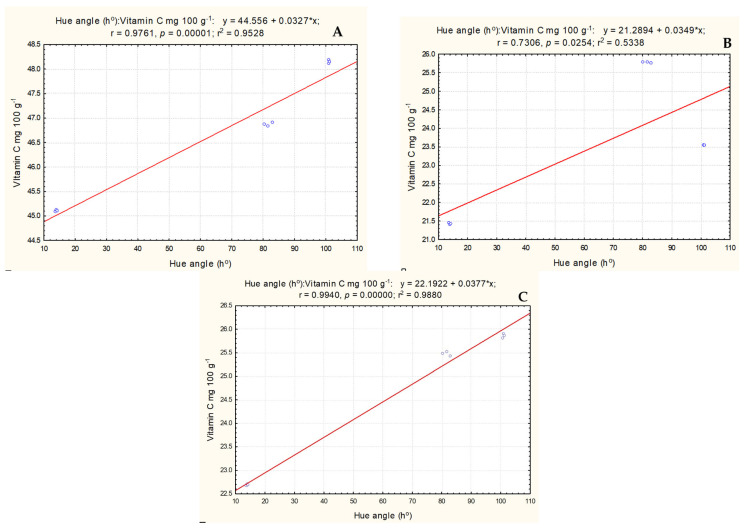
Correlations between hue angle (h°) and vitamin C content. (**A**) *S. melanocerasum**—***SM, (**B**) *S. retroflexum*—SR, (**C**) *S. nigrum*—SN.

**Figure 7 plants-13-00343-f007:**
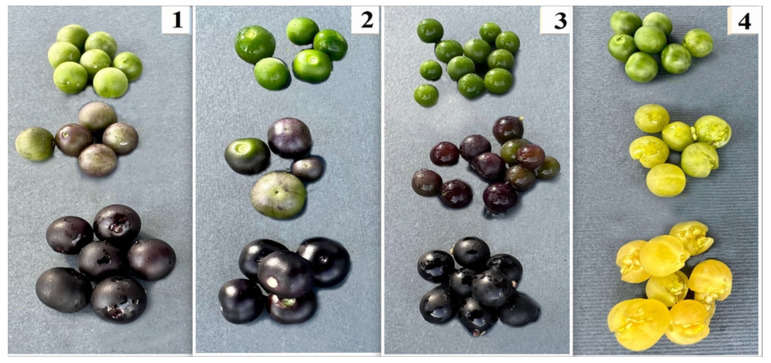
Fruit ripening stages of *Solanum* sp. fruit (photos by J. Staveckienė). 1—*S. retroflexum*, 2—*S. melanocerasum*, 3—*S. nigrum*, 4—*S. vilosum*.

**Table 1 plants-13-00343-t001:** Total mineral element content (mg kg^−1^ DW) in *Solanum* fruits (average for 2021–2022 years).

Microelement				
Ripening Stage/Species	SM	SR	SN	SV
I	375.47 e	519.34 b	374.76 e	411.88 d
II	386.06 e	299.91 f	423.16 d	305.32 f
III	211.12 g	422.82 d	491.10 c	756.48 a
Macroelement				
	SM	SR	SN	SV
I	60,518.65 c	67,035.23 b	50,103.97 e	46,405.89 f
II	37,017.53 i	38,352.13 h	74,763.55 a	26,104.95 l
III	29,171.17 k	58,557.06 d	36,529.45 j	45,750.3 g

Note: Different small letters (a–l) represent significant differences between the means (*p* < 0.05); *S. retroflexum*—SR, *S. melanocerasum—*SM, *S. nigrum*—SN, *S. villosum*—SV; ripening stages I, II, and III.

**Table 2 plants-13-00343-t002:** The color parameter changes of *Solanum* fruits at ripening stage (average for 2021–2022 years).

Ripening Stage	Color Parameter				
	L*	a*	b*	C	h°
	*S. retroflexum*				
I	51.39	−5.90	32.69	33.22	100.23
	51.63	−5.20	32.42	32.83	99.11
	51.63	−5.20	32.39	32.80	99.12
II	49.04	3.15	8.84	9.38	70.39
	50.57	3.15	7.87	8.48	68.19
	50.93	3.21	6.65	7.38	64.23
III	42.44	4.99	4.02	6.41	38.86
	44.40	5.37	2.63	5.98	26.09
	44.82	5.43	2.41	5.94	23.93
*S. melanocerasum*					
I	55.14	−5.23	26.99	27.49	100.97
	54.80	−5.33	27.00	27.52	101.17
	54.67	−5.47	27.15	27.70	101.39
II	43.96	1.27	10.28	10.36	82.96
	43.86	1.46	10.06	10.17	81.74
	43.53	1.61	9.44	9.58	80.32
III	30.30	8.93	2.19	9.19	13.78
	30.27	8.91	2.25	9.19	14.17
	29.96	8.92	2.30	9.21	14.46
*S. nigrum*					
I	46.60	−1.60	21.93	21.99	94.17
	47.14	−1.45	22.09	22.14	93.76
	46.76	−1.60	21.85	21.91	94.19
II	44.05	4.81	5.99	7.68	51.24
	44.03	4.76	6.01	7.67	51.62
	44.11	4.86	5.64	7.45	49.25
III	40.33	6.70	1.47	6.86	12.37
	42.91	7.46	1.24	7.56	9.44
	42.79	7.52	1.15	7.61	8.69

## Data Availability

Data are contained within the article.

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
