# Peer review of "Effects of Different Ripening Stages on the Content of the Mineral Elements and Vitamin C of the Fruit Extracts of Solanum Species: S. melanocerasum, S. nigrum, S. villosum, and S. retroflexum"

_plants, 2024, doi:10.3390/plants13030343_

Round 1

Reviewer 1 Report

Comments and Suggestions for Authors

Dear authors, the paper you propose is quite interesting as it well highlights the variations in the mineral content of fruits of certain Solanum varieties during the ripening process. Nevertheless, some data are difficult to understand, as they are far outside the average values found for individual species: for example, the P content in S. nigrum in phase II, or similarly the calcium content in the same species in phase III. I therefore suggest a careful review of the data to verify that these variations are not due to bias in the analysis process; furthermore, I recommend carefully checking the histogram graphs, as the green and yellow bars both refer to phase II.

Author Response

1

Replies to Reviewers’ comments

Journal name: Plants

Manuscript ID: plants-2731247

Title: Effect of Different Ripening Stages on the Content of the Mineral Elements and Vitamin C of the Fruit Extracts of Solanum Species

Dear Editor,

I’m pleased to submit the revised manuscript after addressing all Reviewers’ comments. All adopted changes are marked in the manuscript with red font. Below you can find our replies to each of the Reviewers’ comments/suggestions/questions

Reviewer no. 1

Point 1: Dear authors, the paper you propose is quite interesting as it well highlights the variations in the mineral content of fruits of certain Solanum varieties during the ripening process. Nevertheless, some data are difficult to understand, as they are far outside the average values found for individual species: for example, the P content in S. nigrum in phase II, or similarly the calcium content in the same species in phase III. I therefore suggest a careful review of the data to verify that these variations are not due to bias in the analysis process; furthermore, I recommend carefully checking the histogram graphs, as the green and yellow bars both refer to phase II.

Response 1: Thank you for your comment, we have corrected the numeration of the ripening stage in Figure 1.  and Figure 2. Thank you for your observation regarding the accuracy of the data. We have checked the data for the increase of P and Ca in the II ripening stage and we want to assure you that these results are correct. Analysis was performed in triplicate, and results are reported as the number obtained during analysis. Such a large increase of these macroelements in the II ripening stage could be due to the stress experienced by the plant. Also, while working with the Solanum fruits, we noticed a tendency that some phytochemical compounds have a sudden increase in the second stage of ripening and a sudden degradation in the third.

Reviewer 2 Report

Comments and Suggestions for Authors

Effect of Different Ripening Stages on the Content of the Mineral Elements and Vitamin C of the Fruit Extracts of Solanum Species

1.    The composition of Solanum species is quite well described in the literature:

Mohyuddin, A., Kurniawan, T. A., Khan, Z. U. D., Nadeem, S., Javed, M., Dera, A. A., ... & Saeed, S. (2022). Comparative insights into the antimicrobial, antioxidant, and nutritional potential of the Solanum Nigrum complex. Processes, 10(8), 1455.

Okeke Philomena, N., Ilodibia Chinyere, V., Ngene Anita, C., & Iroka Finian, C. Anatomical and Biochemical Studies of Solanum melongena and Solanum nigrum.

Zahara, K., Ahmad, N., Bibi, Y., Bibi, F., Sadaf, H. M., & Sardar, N. (2019). An insight to therapeutic potential and phytochemical profile of Solanum villosum (L). Medicine in Drug Discovery, 2, 100007.

Mabotja, T. C. (2019). Effects of irrigation interval and planting density on biomass yield and chemical composition of nightshade (solanum retroflexum) in Limpopo Province, South Africa (Doctoral dissertation).

Moyo, S. M., & Kayitesi, E. (2022). African Nightshade (Solanum nigrum complex species). In Handbook of Phytonutrients in Indigenous Fruits and Vegetables (pp. 97-117). GB: CABI.

Due to the cited literature, the purpose of the work should be better justified. The innovative nature of this project should also be emphasized.

2.    One of the two determinations made in the study is the determination of vitamin C content. It seems that since this is the basic determination, both ascorbic acid and dehydroascorbic acid should be determined. The best method here would be HPLC.

3.    What was the purpose of the color test?

Author Response

2

Replies to Reviewers’ comments

Journal name: Plants

Manuscript ID: plants-2731247

Title: Effect of Different Ripening Stages on the Content of the Mineral Elements and Vitamin C of the Fruit Extracts of Solanum Species

Dear Editor,

I’m pleased to submit the revised manuscript after addressing all Reviewers’ comments. All adopted changes are marked in the manuscript in blue font. Below you can find our replies to each of the Reviewers’ comments/suggestions/questions

Reviewer no. 2

       Point 1.  The composition of Solanum species is quite well described in the literature:

Mohyuddin, A., Kurniawan, T. A., Khan, Z. U. D., Nadeem, S., Javed, M., Dera, A. A., ... & Saeed, S. (2022). Comparative insights into the antimicrobial, antioxidant, and nutritional potential of the Solanum Nigrum complex. Processes, 10(8), 1455.

Okeke Philomena, N., Ilodibia Chinyere, V., Ngene Anita, C., & Iroka Finian, C. Anatomical and Biochemical Studies of Solanum melongena and Solanum nigrum.

Zahara, K., Ahmad, N., Bibi, Y., Bibi, F., Sadaf, H. M., & Sardar, N. (2019). An insight to therapeutic potential and phytochemical profile of Solanum villosum (L). Medicine in Drug Discovery, 2, 100007.

Mabotja, T. C. (2019). Effects of irrigation interval and planting density on biomass yield and chemical composition of nightshade (solanum retroflexum) in Limpopo Province, South Africa (Doctoral dissertation).

Moyo, S. M., & Kayitesi, E. (2022). African Nightshade (Solanum nigrum complex species). In Handbook of Phytonutrients in Indigenous Fruits and Vegetables (pp. 97-117). GB: CABI.

Due to the cited literature, the purpose of the work should be better justified. The innovative nature of this project should also be emphasized.

Response 1: Thank you for your observation. We made some corrections in literature literature purpose.

Point 2.    One of the two determinations made in the study is the determination of vitamin C content. It seems that since this is the basic determination, both ascorbic acid and dehydroascorbic acid should be determined. The best method here would be HPLC.

Response 2: Thank you for your suggestion. Due to the rapid degradation of vitamin C, analyses were performed immediately after harvest and done with a spectrophotometer. In the future, we plan to perform vitamin C analyses with HPLC and compare the results

Point 3.    What was the purpose of the color test?

Response 3: Thank you for your question. The purpose of the color test was to relate the color changes to the stage of ripening and whether it is possible to judge from the color about changes in the mineral element content or vitamin C content depending on the color

Reviewer 3 Report

Comments and Suggestions for Authors

Review Plants 2731247

Effect of Different Ripening Stages on the Content of the Mineral Elements and Vitamin C of the Fruit Extracts of Solanum Species

A well-done paper with a lot of important data that completes the scientific background with potential for publication.

It should be mentioned in the title that the analyses are on some Solanum species - needs to be redone

No abbreviations or abbreviations in the abstract, especially for species, possibly genus names

Lines 48-49 Solanum must be written italic

Line 191 - attention to correct spelling of chemical elements, e.g. Al, Zn

Line 320-326 fruit maturity must use BBCH phenological stage

According with point 3.7 Statistical analyses you used LSD to determine the significance of differences or in graphs you used letters, which means you used Tukey or Duncan. What do the letters a in the graphs represent?

Lines 355-357 P2O5 - be careful when writing, numbers are written as indicators

Write correction mg kg-1

In the conclusions, values should be avoided as much as possible

It does not follow from the conclusions and discussions what is the significance of the fact that some Solanum species have high mineral or vit C content.

What insights can be drawn from these data

Comments on the Quality of English Language

minor changes

Author Response

3

Replies to Reviewers’ comments

Journal name: Plants

Manuscript ID: plants-2731247

Title: Effect of Different Ripening Stages on the Content of the Mineral Elements and Vitamin C of the Fruit Extracts of Solanum Species

Dear Editor,

I’m pleased to submit the revised manuscript after addressing all Reviewers’ comments. All adopted changes are marked in the manuscript in green font. Below you can find our replies to each of the Reviewers’ comments/suggestions/questions

Reviewer no. 3

Point 1: It should be mentioned in the title that the analyses are on some Solanum species - needs to be redone

Response 1: Thank you for your recommendation. We made correction in the title

Point 2. No abbreviations or abbreviations in the abstract, especially for species, possibly genus names

Response 2: Thank you for your observation. We make corrections in line 14-15. And genus name is the same for all species: Solanum

Point 3. Lines 48-49 Solanum must be written italic

Response 3: Thank you for your observation we make corrections

Point 4: Line 191 - attention to correct spelling of chemical elements, e.g. Al, Zn

Response 4: Thank you for your recommendation.Abbreviations of the micro and macro elements  are given in the text, we write the full names under the table, as it is indicated under the graphs. (A) – boron, (B) – natrium, (C) – aliuminium , (D) – manganese, (E) – iron, (F) – copper, (G) – zinc. A capital letter, for example (A), denotes a figure, not elements name.

Point 5: Line 320-326 fruit maturity must use BBCH phenological stage

Response 5: Thank you for your recommendation. We made corrections in  line 340-344

Point 6: According with point 3.7 Statistical analyses you used LSD to determine the significance of differences or in graphs you used letters, which means you used Tukey or Duncan. What do the letters a in the graphs represent?

Response 6: Thank you for your question.  Different small letters (a, b, c, d) represent significant differences between the means (p < 0.05). Explanation written in line 132-133, 193-194. The statistical significance of the differences between the means was estimated using Fisher’s LSD test, line 396. Fisher's least significant difference (LSD) procedure is a two-step testing procedure for pairwise comparisons of several treatment groups

Point 7: Lines 355-357 P2O5 - be careful when writing, numbers are written as indicators

Response 7: Thank you for your observation we make corrections

Point 8: Write correction mg kg-1

Response 8: Thank you for your recommendation we make corrections.

Point 9: In the conclusions, values should be avoided as much as possible

Response 9: Thank you for your recommendation we make corrections.

Point 10: It does not follow from the conclusions and discussions what is the significance of the fact that some Solanum species have high mineral or vitamin C content.

Response 10: Thank you for your observation. We make some explanation in line 400-405

Point 11: What insights can be drawn from these data

Response 11:  Our study confirms that the fruits  Solanum species are a good source of mineral elements and vitamin C. These data are valuable for choosing a ripening stage with the highest accumulation of mineral elements and vitamin C in fruits. Therefore, fruits from Solanum species are encouraged for use in food, cosmetics, and pharmaceuticals. Understanding which ripening stage contains the highest amount of mineral elements or vitamin C can help in pharmaceuticals or improve nutrition

Round 2

Reviewer 2 Report

Comments and Suggestions for Authors

Unfortunately, the answers provided are not satisfactory to me. The authors did not sufficiently justify the methodology of their research. I still believe that methodological changes should be introduced.

Author Response

2

Replies to Reviewers’ comments

Journal name: Plants

Manuscript ID: plants-2731247

Title: Effect of Different Ripening Stages on the Content of the Mineral Elements and Vitamin C of the Fruit Extracts of Solanum Species

Dear Editor,

I’m pleased to submit the revised manuscript after addressing Reviewer no 2

Reviewer no. 2

       Point 1.  Unfortunately, the answers provided are not satisfactory to me. The authors did not sufficiently justify the methodology of their research. I still believe that methodological changes should be introduced.

Response 1: Dear reviewer. We made some changes in methodology during round 1.

  • We made some corrections for the color test according to the BBCH scale. I am uploading a paragraph with corrections for you

Field Experiment

A two-factor experiment with Solanum species: (SM)—S. retroflexum, (SN)—S. nigrum, (SV)—S. villosum, and (SM)—S. melanocerasum, and three different ripening stages: I—ripening stage, fruit color green (30% maturity) corresponding to BBCH stage 81; ripening stage II, fruit color 40–60% purplish-violet or yellow-orange (60% maturity), inside incompletely ripe, corresponding to BBCH stage 85 ; and III−ripening stage, fruit color 100% velvety black-blue or orange, inside fully ripe (100% maturity) corresponding to BBCH stage 89 (Figure 4) [41,42] was conducted during the period 2021–2022 on a farm in the Kaunas district, Lithuania (WGS coordinates 54.8719020, 23.8672686).

  • Also we change the conclusion

Conclusion

     Overall, our findings showed that the amount of microelements varied between 756.48 mg kg-1 DW in SV fruits at III ripening stage and 211.12 mg kg-1 DW in SM fruits at III ripening stage, depending on the ripening stage and species. SV fruits at ripening stage II had a total macroelement concentration of 26.104,95 mg kg-1 DW, while SR fruits at ripening stage I had a total macroelement content of 67.035,23 mg kg-1 DW. SM fruits had the maximum vitamin C content, ranging from 48.15 mg 100g-1 at the I ripening stage to 45.10 mg 100g-1 in the III ripening stage.

    The results of our investigation verify that the fruits of the studied Solanum species are a good source of vitamin C and mineral components. These statistics are useful for selecting a fruit's ripening stage that will result in the highest concentration of mineral elements and vitamin C. Consequently, it is recommended that fruits from the Solanum be used in novel food, cosmetics, and pharmaceutical products. More research is necessary to comprehend the mechanism underlying how the ripening stage of fruit affects quality

Our results showed that the significantly highest amount of microelements was in SV fruits at the III stage of ripening, and the significantly highest amount of macroelements was in SR fruits of the I ripening stage. Significantly highest amount of vitamin C was determined in the SM fruits in the I ripening stage.

We cannot perform vitamin C with HPLC and decompose vitamin C into dehydroascorbic acid and l-ascorbic acid because we do not have the standards and columns for that purpose, and we don't have the foundation to order this analysis or buy columns and standards. The analysis presented in the article is 2-year averages, i.e. 2021-2022. and we doubt that accurate analysis could be repeated after such a time. From your comment, we don't understand exactly what changes you want us to make in the methodological part, and in design appropriate. We would be very grateful if you could detail exactly what changes you want us to make

Round 3

Reviewer 2 Report

Comments and Suggestions for Authors

I conclude that the authors' answers are unsatisfactory.